# Landscape of Immunotherapy Options for Colorectal Cancer: Current Knowledge and Future Perspectives beyond Immune Checkpoint Blockade

**DOI:** 10.3390/life12020229

**Published:** 2022-02-02

**Authors:** Alecsandra Gorzo, Diana Galos, Simona Ruxandra Volovat, Cristian Virgil Lungulescu, Claudia Burz, Daniel Sur

**Affiliations:** 1Department of Medical Oncology, The Oncology Institute “Prof. Dr. Ion Chiricuţă”, 400015 Cluj-Napoca, Romania; alecsandra.gorzo@gmail.com (A.G.); diagalos94@gmail.com (D.G.); cburz@yahoo.fr (C.B.); 2Department of Medical Oncology, University of Medicine and Pharmacy “Iuliu Hatieganu”, 400000 Cluj-Napoca, Romania; 3Department of Medical Oncology, University of Medicine and Pharmacy “Grigore T. Popa” Iasi, 700115 Iasi, Romania; simonavolovat@gmail.com; 4Department of Medical Oncology, University of Medicine and Pharmacy Craiova, 200349 Craiova, Romania; 5Department of Allergology and Immunology, University of Medicine and Pharmacy “Iuliu Hatieganu”, 400000 Cluj-Napoca, Romania

**Keywords:** immunotherapy, colorectal cancer, dMMR, MSI-H, nivolumab, ipilimumab, pembrolizumab, resistance

## Abstract

Colorectal cancer is the third most prevalent malignancy in Western countries and a major cause of death despite recent improvements in screening programs and early detection methods. In the last decade, a growing effort has been put into better understanding how the immune system interacts with cancer cells. Even if treatments with immune checkpoint inhibitors (anti-PD1, anti-PD-L1, anti-CTLA4) were proven effective for several cancer types, the benefit for colorectal cancer patients is still limited. However, a subset of patients with deficient mismatch repair (dMMR)/microsatellite-instability-high (MSI-H) metastatic colorectal cancer has been observed to have a prolonged benefit to immune checkpoint inhibitors. As a result, pembrolizumab and nivolumab +/− ipilimumab recently obtained the Food and Drug Administration approval. This review aims to highlight the body of knowledge on immunotherapy in the colorectal cancer setting, discussing the potential mechanisms of resistance and future strategies to extend its use.

## 1. Introduction

Colorectal cancer (CRC) is the third most prevalent malignancy in Western countries and still a major cause of cancer-related death worldwide [1]. Even if high-income nations show a greater incidence in CRC, less developed countries are encountering significant increases in CRC cases [2]. Extensive studies have observed differences in risk factors, incidence and cancer-related deaths between ethnic groups, with African Americans, in particular, showing a higher frequency of CRC cases as well as death rates [3,4,5]. Although sustained efforts have been made in improving screening and early detection approaches, about 25% of the CRC patients are diagnosed with metastatic disease and, therefore, have a very poor prognosis [1]. With drug combination optimization, mortality has been reduced; however, the five-year overall survival (OS) remains only 20% [6].

In early-stage CRC, surgery represents the elective treatment [7]. The five-year survival rate following surgical resection is 99% in stage I, 68–83% in stage II, and 45–65% in stage III [8,9]. Therefore, adjuvant chemotherapy is administered in stage III and high-risk stage II (T4 stage, less than 12 lymph nodes examined, positive resection margins, lymphovascular emboli, perineural invasion, obstruction, and perforation) CRC patients to prolong overall survival. The standard of care in the adjuvant setting is the combination of fluoropyrimidine and oxaliplatin (FOLFOX or CAPOX), which was shown to significantly improve disease-free survival (DFS), compared to fluoropyrimidine alone [10]. Moreover, the addition of oxaliplatin resulted in improved OS and reduced risk of death with 17%, 16%, and 12% in the studies conducted by XELODA, MOSAIC, and NSABP C-07, respectively [11,12,13].

In the metastatic setting (mCRC), the median OS is approximately 30 months. In selected patients, surgical resection of metastases is advisable either upfront or following downsizing systemic treatment [14,15]. If surgical resection does not represent a realistic goal, systemic therapy with chemotherapy and targeted agents has been shown to substantially increase overall survival [16]. Standard chemotherapies are represented by fluoropyrimidines, oxaliplatin, irinotecan, and trifluridine/tripiracil [17]. The efficacy of chemotherapy agents is further improved by the addition of targeted agents, such as the anti-EGFR mAbs (cetuximab and panitumumab) in RAS-wt tumor or anti-VEGF agent bevacizumab, regardless of RAS status [6,18,19]. Ramucirumab (anti-VEGFR-2) and aflibercept (a synthetic receptor for VEGF-A, VEGF-B, and PIGF) have demonstrated efficacy in the second-line setting, in combination with chemotherapy [20,21]. Regorafenib, a multikinase inhibitor, has also demonstrated efficacy in further lines such as monotherapy [22].

Throughout the last decade, the immune system was deeply studied to understand how it interacts with cancer cells. Immune checkpoint inhibitors (ICIs) manipulate the immune system to reactivate the antitumor immune response by blocking the immune checkpoint proteins (PD-1 and CTLA-4) or their ligands (PD-L1). Consequently, ICIs, such as anti-PD-L1 monoclonal antibodies (mAbs-Atezolizumab, Avelumab, and Durvalumab), anti-PD-1 mAbs (Nivolumab and Pembrolizumab), and anti-CTLA-4 mAbs (Ipilimumab) led to marked therapeutic efficacy in melanoma, as well as lung, head and neck, and urothelial cancers [23,24,25].

Furthermore, to correlate the phenotype of cancer cells with the clinical behavior and guide treatment, CRC has been classified into four consensus molecular subtypes (CMS): CMS1 (14%)—MSI Immune, with strong immune activation, hypermutated, MS unstable; CMS2 (37%)—canonical, epithelial, chromosomally unstable, with marked WNT and MYC signaling activation; CMS3 (13%)—metabolic, epithelial with metabolic dysregulation; and CMS4 (23%)—mesenchymal, prominent transforming growth factor-beta activation, stromal invasion, and angiogenesis, with a remaining unclassified group (13%) with mixed features [26]. Recent studies have confirmed an increased response rate to ICIs in patients whose tumors are highly microsatellite unstable (MSI-H) and are DNA mismatch repair-deficient (dMMR) [27,28]. This subset of patients with this unique phenotype represents about 15% of all sporadic CRC and only 5% of mCRC [29]. To date, based on the result of phase III KEYNOTE-177 and phase II Checkmate-142 trial, the NCCN guideline recommends the use of pembrolizumab and nivolumab ± ipilimumbab in the first-line and non-first-line settings in MSI-H/dMMR mCRC patients. Moreover, the checkpoint inhibitors regimens are also recommended in the neoadjuvant setting for resectable mCRC with MSH-I/dMMR status [30,31].

This article aims to review the existing data for applying immunotherapy in CRC, more precisely in a subset of patients with MSI-H tumors. We addressed the importance of biomarkers in selecting CRC patients for immunotherapy. The review also discusses the challenges due to resistance mechanisms and potential future strategies to extend immunotherapy uses.

## 2. Predictive/Prognostic Biomarkers for Selecting CRC for Immunotherapy

### 2.1. Microsatellite Status

Microsatellites are repeated non-coding DNA sequences, ranging in length from 1 to 10 base pairs. During DNA replication, microsatellites are frequent sites for mutations [32]. The role in detecting and correcting these errors is assigned to the mismatch-repair system (MMR) [33]. The presence of microsatellite instability (MSI-H) due to deficiency in the MMR (dMMR) is found in about 15% of all CRCs and 4% of mCRC cases [34]. MSI-H status is the hallmark of Lynch syndrome; however, 70–85% of patients with MSI-H/dMMR tumors have somatic mutations, most frequently inactivating the MLH1 gene [35]. The MSI status can be detected either by polymerized chain reaction (PCR), next-generation sequencing (NGS), or by the absence of immunohistochemical staining of the MMR proteins (MSH2, MLH1, MSH6, and PMS2) [36].

This phenotype is characterized by a widespread accumulation of mutations, which generate frame-shifted proteins (neoantigens) with great immunogenic potential. MSI-H/dMMR tumors frequently involve the proximal colon, are poorly differentiated, and have mucinous histology [37]. Population-based studies have investigated the susceptibility of MSI-H/dMMR mutations in certain ethnic groups presenting with CRC. Studies show the MSI-H/dMMR phenotype has a significantly higher prevalence in the African American (AA) population (up to 45%) [38,39,40], while the Caucasian and Asian populations show a lower incidence of MSI-H rates, no higher than 20% [41,42]. A study conducted on an Indian cohort observed similar frequencies of MSI-H/dMMR CRC in their studied population when compared to the West, despite having an inferior incidence of CRC cases [43]. MSI-H/dMMR tumors are highly infiltrated with immune cells, including CD4+ TILs (tumor-infiltrating lymphocytes), CD8+ TILs, Th1 (T helper 1), and macrophages [37]. Moreover, these tumors have an up-regulated expression of immune checkpoints (PD-1, PD-L1, CTLA4) [44]. Therefore, based on these observations, it was suggested that MSI-H/dMMR CRCs might have a good response to ICIs. Following the results of several stage II trials, ICIs are considered a breakthrough strategy in the treatment of MSI-H/dMMR mCRC [45,46]. However, not all MSI-H/dMMR mCRC patients respond to ICIs, suggesting that a deeper knowledge of immune-related mechanisms is needed [47]. In stage II CRC, MSI-H/dMMR is associated with a lower recurrence rate than MSI-L/pMMR tumors, with an HR estimated for OS correlated with MSI of 0.65 (95% CI: 0.59–0.71) [48].

### 2.2. Tumor Mutational Burden

The success linked to ICIs was associated with the hypermutated phenotype due to many DNA replication errors, alongside the consequent inflamed TME [49]. Hence, an emerging biomarker to predict the tumor’s response to immunotherapy is the calculation of tumor mutational burden (TMB), which quantifies the nonsynonymous mutations per coding area in a tumor genome [50]. The relationship between immunotherapy response and TMB could be explained by the fact that a higher number of mutations generates higher mutation-associated neoantigens, with increased lymphocyte infiltration in the TME [51]. The Food and Drug Administration (FDA) approved treatment with Pembrolizumab in patients with any refractory and unresectable or metastatic solid tumors that harbor a high TMB (TMB-H), defined as ≥10 mutations per megabase (Mut/Mb) [52]. The decision was based on the analysis of 10 cohorts of patients with metastatic solid tumors enrolled in the KEYNOTE-158 trial, which investigated the treatment with pembrolizumab until disease progression or unacceptable toxicities occurred. Among all patients (*n* = 790), 13% (*n* = 102) were defined as TMB-H. The results showed a 29% RR in TMB-H patients, compared to 6% in those with TMB <10 mut/Mb [53]. The relationship between TMB and other clinicopathologic variables that are already known to influence immunotherapy response remains to be elucidated.

### 2.3. Immunoscore

Increasing evidence has shown that cancer evolution is strongly dependent on the TME consisting of various cell entities, including blood vessels, endothelial cells, fibroblasts, and cells of the immune system. It was demonstrated that adaptive immune cell infiltration is a better prognostic marker than grading, staging, and metastatic status [54,55]. The immunoscore is a digital pathology and immunohistochemistry-based assay which translates the immune contexture into a feasible prognostic biomarker for stage I-III CRC [56]. The immunoscore summarizes the density of the lymphocyte population, CD3+ and CD8+ T cells in the tumor core, and invasive margins, which provides a scoring system from low immune cell density (immunoscore 0) to high density (immunoscore 4). At the center of this mechanism, the more a tumor is defined as immunogenic, the more it is able to attract a T-cell mediated immune response, which is associated with a higher neoantigen load, often found in MSI-H/dMMR tumors [57]. Consequently, a high immunoscore correlates with a longer patient’s survival [58], while patients with low immunoscore and minimal tumor invasion are more likely to undergo disease relapse [59,60]. In terms of prognostic ability, the immunoscore tends to outperform the classical gold-standard TNM system in predicting DFS and OS for stage I, II, III CRC [61,62]. A high immunocore was associated with the highest DFS and OS in stage II colon cancer. The five-year recurrence rate was 8% in high, 14% in intermediate, and 23% in low immunoscore. A multivariable analysis had similar results (*p* < 0.0001 for high Immunoscore vs. low) [63]. Regarding stage III colon cancer, according to the NCCTG NO157 trial, a high immunoscore was correlated with a longer three-year DFS compared to a low immunoscore (*p* < 0.05) [64]. In the phase III IDEA trial, high and intermediate immunoscore significantly predict a DFS benefit of prolonged adjuvant chemotherapy with FOLFOX regimen in stage III colon cancer patients. (HR = 0.53; 95% CI 0.37–0.75; *p* = 0.0004) [65]. Apart from representing a good prognostic marker, the immune contexture could also predict the response to ICIs [66]. The CD8+ T cells’ density was directly correlated with the clinical response to anti-PD1 agents. Moreover, CD8+ T cells were also suggested to be a good predictor of the response to anti CTLA4 molecules in melanoma patients [67]. The validation of the consensus immunoscore [58] and its introduction in the fifth edition of the World Health Organization classification of digestive tumors among the “Essential and Desirable Diagnostic Criteria” for CRC [68] makes it a step closer to the proposed notion of TNM-I classification (“I” from “immune”) [69].

### 2.4. POLD1/POLE

DNA polymerase delta (POLD1) and DNA polymerase epsilon (POLE) are two key enzymes responsible for the accurate replication of the genome in the cell cycle. Mutations that occur in the POLE and POLD1 genes generate a deficit in DNA repair [44]. Therefore, they lead to an ultra-mutated phenotype, with up to 10 times more mutations than in MSI-H CRC. Germline mutations in the exonuclease domain of POLD1 and POLE affect the proofreading abilities of these polymerases, predispose to multiple colorectal carcinomas and adenomas, and generate polymerase proofreading-associated polyposis (PPAP) [70]. PPAP represents 0.1–0.4% of familial cancer cases [71]. Moreover, other extracolonic tumors were described, including brain, endometrial, ovarian, breast, skin tumors [72]. Most of these mutations represent, however, somatic events [73]. This type of tumor has similar characteristics with the dMMR CRC, including up-regulation of immune checkpoint molecule, high level of TILs, and increased cytotoxic T cells markers [74]. Clinically, POLE-mutated CRC patients usually have a good prognosis. The tumors are characterized by an early stage at presentation, right-sided location, male sex, and younger age [75]. Given the similarities with the MSI-H/dMMR CRCs, the therapeutic potential of ICIs in POLE-mutated CRCs is clinically relevant. To date, very few scientific works are available about the efficacy of ICIs in mCRC harboring POLE or POLD1 mutations [76,77]. Further data are needed to assess if mutations in POLE and POLD1 might predict benefits from ICIs.

### 2.5. PD-1/PD-L1 Expression

The PD-1 molecules are expressed by activated NK cells, B-cells, and T-cells, and they can bind to their ligands, PD-L1, expressed on cancerous cells [78]. One of the most extensively studied biomarkers is probably the tumor expression of PD-L1 determined by immunohistochemistry [79]. Even if in esophageal, gastric, and NSCLC the PD-L1 expression could be a valuable predictor of response to anti-PD-1 therapy [79,80], it was not formally demonstrated to be associated with survival or response to immunotherapy in CRC [81,82]. Several issues are preventing the expression of PD-L1 from being a reliable biomarker. Firstly, PD-L1 expression represents a dynamic process that adapts according to the tumor stage and microenvironment, and it can also be influenced by treatment. The tumor expression is not uniform; therefore, the sampling location and time could affect the results of PD-L1 staining [83]. Nonetheless, lacking a standard evaluation for PD-L1 expression limits its clinical significance [84].

## 3. Immunotherapy in CRC

One of the strategies that have been revolutionizing cancer treatment in the last few decades revolves around targeting the immune system. Immunotherapy aims at overcoming the limitations of chemotherapy and radiotherapy while targeting the host’s own immune system. Once administered, immunotherapy alerts the innate and adaptive immune responses about the presence of cancerous cells and guides the immune response toward eradicating them, leaving healthy cells unaffected [85]. These drugs can be administered as passive immunotherapy (immunostimulatory cytokines, immunomodulatory mAbs, dendritic cell-based immunotherapies, anti-cancer vaccines, inhibitors of immunosuppressive metabolism, pattern recognition receptors, and immunogenic cell-death inducers) or as active immunotherapy (adoptive cell transfer, oncolytic viruses, or tumor-targeting mAbs), with some of these strategies finding their utility in CRC treatment [86].

### 3.1. Immunomodulatory mAbs

Immune checkpoints represent a set of regulatory pathways of the immune system whose primary role is to ensure modulation and control of the immune response while maintaining self-tolerance [87]. Two such pathways are PD-1/PD-L1/2 and CTLA-4/CD80-CD86 and they represent encouraging targets for immunotherapies. These molecules are present on tumor cells, T cells, and antigen-presenting cells (APC). Once the co-inhibitory receptor (PD-1, CTLA4) interacts with its ligand (PD-L1/2 and CD80-CD86, respectively), the T-cell function is inhibited, leading to a suppressed immune response [88,89]. Cancerous cells exploit this mechanism by hyperactivating immune checkpoints and overexpressing ligands; therefore, evading the immune response. ICIs attempt to dampen the PD-1/PD-L1/1 and CTLA-4/CD80-CD86 interaction, restore immunosurveillance and aid the host’s immune system in fighting cancer [90].

ICIs targeting PD-1 and CTLA4 have demonstrated significant activity in solid tumors, such as melanomas, non-small-cell lung cancer (NSCLC), and renal cell carcinoma [91]. The initial studies showed limited clinical activity of ICIs in non-selected CRC patients. A phase I trial (NCT00730639) investigating the role of nivolumab (anti-PD-1) in advanced solid tumors, including 19 CRC patients, reported a complete response that lasted over three years in just one of the CRC patients with an MSI-H/dMMR phenotype [92,93]. Based on these results and understanding of tumor microenvironment in MSI-H/dMMR tumors, the interest in immunotherapy in CRC was further expanded. The mechanism behind the ICIs is depicted in Figure 1.

Pembrolizumab is a PD-1 blocking humanized, IgG4 monoclonal antibody which prevents the interaction between PD-1 and its ligands, PD-L1, and PD-L2 [94]. The phase II trial, KEYNOTE-028, investigated the clinical activity of pembrolizumab (10 mg/kg) in MSI-H tumors. Of the 41 patients included, 32 cases had mCRC with or without MSI-H/dMMR phenotype. Among the MSI-H/dMMR mCRC patients, the objective response rate (ORR) was 40%, and a disease-control rate >12 weeks was observed in 90% of cases. The possibility of achieving a response to therapy was significantly associated with the number of somatic mutations (*p* = 0.02) [27]. Considering these outcomes, in May 2017, the FDA approved pembrolizumab to treat MSI-H/dMMR advanced CRC patients that progressed on conventional chemotherapy [95].

Furthermore, the randomized, open-label phase III study KEYNOTE-177 demonstrated a significant improvement in median progression-free survival (PFS) and ORR after administration of pembrolizumab in MSI-H/dMMR mCRC when compared to 5-fluorouracil-based chemotherapy alone or in combination with bevacizumab or cetuximab, with acceptable toxicity. Median PFS was significantly longer in the pembrolizumab arm (16.5 months), compared to the chemotherapy arm (8.2 months). Confirmed ORR reached 43.8% with pembrolizumab vs. 33.1% with chemotherapy. Based on these data, pembrolizumab was recommended as the first-line treatment for patients with mCRC and MSI-H/dMMR [96].

Nivolumab, a human IgG4 mAbs, is another PD-1 inhibitor approved for mCRC with MSI-H/dMMR [97]. CheckMate-142, a phase II open-label trial, investigated the efficacy of nivolumab (3 mg/kg, every 2 weeks) in 74 previously treated MSI-H/dMMR mCRC patients. The 12-month PFS was reported to be 50% (95% CI: 38–61), and the 12-month OS was 73% (95% CI: 62–82) [98]. Based on these outcomes, in 2017, the FDA approved nivolumab to treat MSI-H/dMMR mCRC with progressive disease after chemotherapy [99]. The trial further evaluated the combination of nivolumab + low-dose ipilimumab (anti-CTLA4 mAb). At a median follow-up of 13.4 months, the results showed an improved clinical benefit with an ORR of 55%. Regardless of the PD-L1 tissue expression, 71% of the patients remain progression-free at 12 months. Moreover, the 12-month median OS was 85% (95% CI: 77.0–90.2) [100]. In 2018, these results led to the FDA approval of nivolumab + low-dose ipilimumab for previously treated MSI-H/dMMR mCRC patients [101].

In addition, the trial further evaluated the role of the combination therapy as first-line treatment for MSI-H/dMMR mCRC. After a median follow-up of 29.0 months, the study showed a significant ORR (69%) and CR (13%), but with OS and median PFS not yet reached [102]. Based on these results, to date, nivolumab is approved as the first-line therapy option for MSI-H/dMMR mCRC, either as monotherapy or in combination with ipilimumab [103].

Avelumab is an anti-PD-L1 inhibitor evaluated in mCRC with MSI-H/dMMR status. A recently conducted phase II study evaluated monotherapy with avelumab in patients with MSI-H/dMMR or POLE-mutated metastatic or unresectable CRC and presented encouraging results with manageable toxicities. The primary endpoint of the study was ORR, which was evaluated at 24.2% overall and 28.6% in patients with MSI-H/dMMR. In terms of secondary endpoints, PFS rates were 3.9 and 8.1 months in the MSI-H/dMMR patients, and the median OS was 13.2 months. The results presented by this trial showed that avelumab efficacy in the mCRC setting is comparable with that of FDA-approved pembrolizumab and nivolumab [104]. Several combination therapies between avelumab and other therapeutical agents are being investigated at the moment [105].

Another ICI regimen is the one combining an anti-PD-L1 agent (durvalumab) with the anti-CTLA-4 drug tremelimumab in mCRC. A randomized phase II study demonstrated prolonged OS with durvalumab+tremelimumab vs. best supportive care (BSC). Moreover, the study analyzed the possibility of using plasma TMB for selecting patients for immunotherapy. Patients with elevated TMB would most likely benefit from durvalumab and tremelimumab combination. After a median follow-up of 15.2 months, the median OS of the combination ICI therapy was 6.6 months, while the median OS for the BSC arm was 4.1 months [106].

Atezolizumab, an anti-PD-L1 mAb, is currently being investigated in the adjuvant CRC setting. The phase III ATOMIC trial compares the combination of atezolizumab and FOLFOLX vs. FOLFOX alone in MSI-H stage III CRC patients. The study has DFS as the primary endpoint and will establish if ICIs might be added to the oxaliplatin-based regime in this setting [107].

### 3.2. Neoadjuvant Setting

Immune checkpoint blockade has also been discussed as a neoadjuvant strategy, although studies have reported only a few cases. One case report had shown significant benefits when pembrolizumab was administered in the neoadjuvant setting in a Lynch syndrome patient, who after that qualified for surgical resection [108]. A retrospective study on two patients with locally advanced CRC has shown that nivolumab in the neoadjuvant setting can induce complete responses, either as a single treatment option or followed by surgery [109]. Moreover, nivolumab has proved to induce a significant pathological response as neoadjuvant treatment for early stage CRC, when administered in combination with ipilimumab, as demonstrated by a different clinical trial (NCT03026140) conducted in Europe [110]. Even if, to date, no clinical trials are supporting this approach, the NCCN guideline recommends the administration of pembrolizumab or nivolumab ± ipilimumab as an option for neoadjuvant setting in resectable MSI-H/dMMR mCRC [111].

### 3.3. Adoptive Cell Transfer

Neoantigens represent an emerging target for immunotherapeutic approaches attempting to overcome the toxicities and narrowed response rates of non-antigen-specific treatments [112]. Neoantigens are altered peptides derived from non-synonymous somatic mutations otherwise absent in normal tissues. These tumor-specific antigens are presented by major histocompatibility complex I (MHC) proteins and then recognized by T-cells and triggering an anti-tumor T-cell immune response [113]. Next-generation sequencing technologies are utilized with the purpose of identifying neoantigens suited to activate tumor-specific T-cell recognition [114].

Adoptive cell transfer (ACT) describes the neoantigen-targeting strategy that requires immune cells derived from patients (autologous transfer), donors (allergenic transfer), or cells differentiated from stem cells. These cells are activated and expanded in vitro through gene modification processes in order to make them better suited to target cancerous cells and eradicate them, thereby improving the immune functions once infused into the patient as therapy [115]. ACT technologies include the manipulation of the host’s tumor-infiltrating lymphocytes (TILs) and the host’s T-cells that have been genetically altered to express a T-cell receptor or a chimeric antigen receptor (CAR) [116]. Research into ACT therapy has objectified clinical responses in the settings of cholangiocarcinoma [117], breast cancer [118], metastatic melanoma [119,120], and CRC [121]. During a phase II clinical trial (NCT01174121) assessing the efficacy of adoptive transfer of autologous TILs in certain solid tumors (digestive tract, urothelial, breast, ovarian, and endometrial cancers), one patient with mCRC showed objective regression. Following one infusion with TILs reactive to KRAS G12D mutation identified in the tumor, the patient presented with regression of all seven lung metastases. Nine months after therapy, one of the seven lesions showed progression, and it was subjected to resection. After the removal of the lung lesion, the patient remained clinically disease-free for four months [121].

CAR-T cell therapy has also been explored in the setting of CRC. CAR-T cells can be manipulated to target a series of tumor-associated antigens (TAAs) highly expressed by CRC tumors, most notably carcinoembryonic antigen CEA [122]. A phase I clinical trial (NCT02349724) indicated that CEA CAR-T cell therapy has some efficacy in mCRC patients with CEA positive tumors, with an acceptable toxicity profile. The authors reported stable disease in 70% of the patients who received infusions with CAR-T cells, while two patients showed tumor regression. In addition, the study observed a sustained decline in the levels of serum CEA [123]. Furthermore, other phase I clinical trials have addressed the efficacy of CAR-T cells targeting CEA as a regional treatment for liver metastases (NCT01373047, NCT02416466) from CRC. The study conducted at Boston University concluded that percutaneous hepatic artery infusions of anti-CEA CAR-T cells showed promising signs of clinical response in patients who underwent multiple lines of systemic therapy, with a safe toxicity profile [124]. The use of anti-CEA CAR-T cell therapy as local treatment of peritoneal carcinomatosis from mCRC has also been studied in pre-clinical trials [125] and several other studies are underway analyzing the further clinical impact of ACT in CRC (NCT03935893, NCT03970382, NCT04426669).

### 3.4. Cancer Vaccines

In mCRC, several types of tumor vaccines were studied, including peptide vaccines, autologous vaccines, dendritic cell transplants, and oncolytic viral vector vaccines, but with limited efficacy [126]. The rationale behind viral antigen vaccines is based on the pathogenicity of the virus, which can generate a robust immune response [127]. Therefore, oncolytic virotherapy demonstrated antitumor efficacy when administered alone, or alongside conventional chemotherapy [128]. Further research has identified potential targets for peptide vaccine-based immunotherapy in TAAs over-expressed on the surface of tumor cells. In the case of CRC, the targeted molecules were CEA, melanoma-associated antigen, and MUC1 [129]. A phase II trial assessed the survival benefit of autologous dendritic cells modified with a pox vector encoding MUC1 and CEA (PANVAC) in mCRC who were disease-free after metastasectomy and perioperative chemotherapy. The survival was longer in the group of patients who received active immunotherapy [130]. Although cancer vaccine TAAs have demonstrated their capacity to strengthen the immune system and presented low toxicities, the evidence showing a reliable survival benefit is limited [131,132,133].

As aforementioned, deficiencies in the MMR proteins generate genomic instability at the sites of microsatellite coding sequences. This phenomenon results in frameshift antigens considered to be highly immunogenic and a good target for vaccines [130]. Therapeutic vaccines targeting tumor-specific neoantigens intend to enhance the existing effector T-cells, expend new antitumor T-cells clones, and contribute to tumor destruction [134]. These vaccines are formulated as RNA or DNA coding for neoantigens, virus-based systems, synthetic peptides, or dendritic cells loaded with neoantigens [135]. After the encouraging results from mouse models, the first-in-human trials investigating neoantigen vaccines demonstrated their safety and efficacy in glioblastoma and melanoma patients [136,137]. To date, neoantigen-based vaccines with or without ICIs have been investigated in various solid tumors, including CRC (NCT04087252, NCT03289962, NCT03639716, NCT0355271). Furthermore, we summarize the current immunotherapeutic options in CRC in Figure 2.

### 3.5. Highlights on Randomized Clinical Trials

Immunotherapy has shown great efficacy in MSI-H/dMMR CRC [102,138]. However, it is still a challenge to identify the optimal line of therapy and possible novel combinations. The ongoing and completed clinical trials investigating anti-PD1, anti-PD-L1, and anti-CTLA4 are listed in Table 1 and Table 2, respectively.

## 4. Resistance to Immunotherapy

Even if the administration of ICIs in MSI-H/dMMR CRC patients is relatively recent, resistance to treatment was already reported. The clinical studies investigating ipilimumab-nivolumab and pembrolizumab showed objective responses of 54.6% [45] and 40% [27], respectively. The results suggest that a group of MSI-H/dMMR CRC patients harbor mechanisms of resistance that impair immune antitumor activity [139]. One of the mechanisms by which cancer cells avoid immune surveillance is altering the expression of the human leukocyte antigen (HLA) complex, leading to inadequate antigen processing and presentation [140]. A study including 179 MSI-H/dMMR CRC patients from the Nurse Health Study, Tumor Cancer Genome Atlas, and the Health Professionals Follow-up Study cohorts investigated the potential immune evasion mechanism [141]. The study described alterations in the immune-response-related genes correlated to B-cells development, T-cells response, and NK cell function. Most of the MSI-H tumors harbored at least one mutation that could impair antigen presentation. Although in, the majority of cases, these initial mutations were not sufficient to confer resistance to ICIs, they suggest that immune editing is preceding the treatment and tumors are on a resistance continuum. β-2 microglobulin (B2M) is known to be an important part of the HLA-class I complex. Mutations in the B2M gene result in the complete loss of HLA class I molecules on the cell surface [142]. Therefore, B2M deficiency was considered a negative prognostic factor in various tumor types and linked to immune escape [143,144]. B2M somatic mutations were found in about 30% of the dMMR CRCs and less than 2% in pMMR tumors [145]. These mutations occur very often in the coding microsatellites as a result of microsatellite instability and were correlated with resistance to anti-PD-1 molecules [146,147]. It is currently unclear if new clones with defect antigen-presenting machinery evolve due to MSI and genomic instability during the immune checkpoint treatment or the selection of preexisting clones with B2M alterations leads to resistance [148].

Tumor-specific antigen expression plays a significant role when talking about the persistence of antitumor immune response. MSI-H/dMMR tumors generate about 50 times more neoantigens than MSS tumors due to frameshift mutations resulting from MMR deficiency [149]. This aspect brings up an important issue regarding the quality of mutations. Point mutations, causing limited amino acid changes in the protein structure, are less likely to generate a solid immune response, compared to mutations affecting the antigenic structure of proteins [150]. For instance, KRAS point mutations are an important step in developing many solid tumors; however, they show poor immunogenic activity [151]. It is essential to mention that the loss of MMR-gene expression might not always represent MSI status. Consequently, patients could present with an MSI-L disease similar to the MMS phenotype, and therefore with inadequate response to immunotherapy [152].

Myeloid-derived suppressor cells (MDSCs) are a heterogeneous population represented by immature myeloid cells with immune regulatory functions in various diseases, such as chronic inflammation, autoimmune diseases, and viral infections [153]. Accumulating evidence suggested that, in cancer-bearing hosts, MDSCs actively contribute to resistance to immunotherapy [154,155]. MDSCs inhibit the activation and cytotoxicity of T cells and were also shown to favor Treg differentiation and expansion [156]. Additionally, they were shown to be involved in an array of non-immunologic processes, including promotion, angiogenesis, and metastasis [157]. In breast cancer models, the accumulation of circulating MDSCs was associated with unresponsiveness to anti-CTLA4/anti-PD-1 [158]. In melanoma patients, low circulating MDSCs levels are common among clinical responders to ipilimumab [159]. In CRC, MSI-L/pMMR tumors were reported to be highly infiltrated with MDSCs and Treg, compared to MSI-H/dMMR, which might explain the poor outcome of ICIs [160]. To better select the population who could most benefit from immune checkpoint inhibitors, further studies are needed to determine if negative regulatory cells should be included in biomarker systems, such as immunoscore [58].

In patients who developed resistance to PD-1 blockade, the whole-exome sequencing of the tumors showed mutations in Janus kinases 1 and 2 (JAK1 and JAK2) [161]. Truncating mutations in JAK 1/2 were correlated to a lack of IFN-γ responsiveness in cancer cells and, consequently, with secondary resistance to ICIs [162,163]. Additionally, in JAK mutated MSI CRCs, melanoma, endometrial cancer, the expression of the PD-L1 gene was significantly down-regulated [164,165]. In melanoma cells, it was shown that the IFN-γ signaling pathway regulates the expression of PD-L1 through JAK1/2. Therefore, cancer cells might evade IFN-γ-immune response throughout JAK1/2 mutations, leading to impaired IFN-y signaling and preventing PD-L1 expression [166]. Signal transducer and activator of transcription proteins (STATs 1/2), members of this pathway, which function downstream of JAK signaling, are potent mediators of IFN-γ. Mutations in STAT proteins resulting in loss of function could generate impaired IFN-γ signaling and, therefore, immune escape [167].

It was recently discovered that the Wnt/β-catenin pathway coordinates the tumor microenvironment and immune cell infiltration [168]. A preclinical study using murine melanoma models demonstrated that increased expression of the Wnt/β-catenin pathway could decrease IFN-γ levels and T-cell function as a consequence [169]. Another study on melanoma cells also showed that hyperactivation of the Wnt/β-catenin pathway reduces T-cell infiltration in the TME, leading to reduced efficacy of ICIs [170]. Regarding MSI-H/dMMR CRC patients, the WNT signaling pathway should be further analyzed in the context of ICIs responsiveness.

## 5. The Future of Immunotherapy in CRC

### 5.1. A New Generation of Immune Checkpoint Inhibitors

The interest over other immune checkpoints increased significantly in the last years, and new potential targets were identified, such as LAG-3, TIM-3, TIGIT, or VISTA [171]. These receptors were shown to be highly expressed on TILs, compared to circulating T-cells found in CRC patients [172]. In several tumors, such as ovarian, melanoma, NSCLC, and gastrointestinal cancers, PD-1 was usually co-expressed with LAG-3, TIM-3, TIGIT, and VISTA on TILs In ovarian cancer, the number of PD-1+LAG-3+CD8+T-cells expressing TNF-α and IFN-γ were significantly decreased, compared to their equivalents without the co-inhibitory receptors [173,174,175,176]. Similarly, in CRC, it was shown that the amount of tumor-infiltrating CD8+ lymphocytes producing IFN-γ was reduced when expressing both TIM-3 and PD-1 [177]. Considering these observations, it might be assumed that using a single anti-PD-1 agent is not always enough to restore T-cells’ functionality. Based on this rationale, many clinical studies investigating new generation checkpoint inhibitors were implemented. V-domain Ig Suppressor of T cell Activation (VISTA) is an immune inhibitory receptor involved in maintaining peripheral tolerance, and it also inhibits the effector function of T cells [178]. Le Mercier et al. demonstrated that VISTA blockade altered the suppressive hallmark of the TME and enhanced specific T-cell response in tumor cells [179]. To date, there is an ongoing phase I clinical study with a fully human mAb anti-VISTA tested in advanced solid tumors (NCT04475523). Preliminary results showed that the administration of the anti-LAG-3 antibody, R3767, led to stable disease in 11 out of 27 patients with advanced solid tumors [180]. Anti-TIM-3 antibodies MBG42 and LY3321367 were well tolerated in monotherapy or when combined with other ICIs [181,182].

Another approach to improve immunotherapy outcomes is to combine ICIs with co-stimulatory checkpoint molecules, such as anti-ICOS, CD28, TNFRSF7, TNFRSF9, and glucocorticoid-induced TNFR-related protein. OX40 antigen (CD134) is a part of the tumor necrosis factor receptors family, and alongside its ligand, OX40L stimulates the activation and proliferation of CD4+ and CD8+ [183]. Various clinical trials are investigating the activity and safety of OX40 agonists in monotherapy or combined with ICIs [184,185].

In order to extend the curative potential of cancer immunotherapies, novel delivery systems are needed. Ongoing research investigates various delivery platforms such as implants, nanoparticles, biomaterials, and scaffolds [186]. Among their many benefits, we can mention the following: protecting and keeping the cargo inactive until it accumulates in the targeted cells, allowing localized and controlled drug delivery to minimize toxicities [187]. For example, to reduce the side effects following systemic administration, ICIs were linked to a peptide derived from PLGF2 (placental growth factor 2) with a good affinity for numerous matrix proteins. In melanoma and breast cancer models, these conjugates remained more localized near the tumor site after peritumoral administration [188]. Moreover, new delivery platforms, such as nanoparticles, could attenuate drug exposure of particular tissues caused by therapeutic combinations (chemotherapy and immunotherapy) that would otherwise be toxic for the patient [187,189]. Besides the many benefits already mentioned, new delivery technologies could address the limitations set by resistance mechanisms. For instance, delivery systems could be expanded to modulate immunogenicity in tumors with cold microenvironments and enhance the response to ICIs [190]. As immunotherapy is evolving very fast, all the advances made in drug delivery will significantly contribute to personalized medicine.

### 5.2. Synergy of Immunotherapy with Other Therapies for MSI-L/pMMR

In most mCRC cases (about 95%) defined as pMMR, ICIs failed to provide clinical benefits due to the immune deserted TME [191]. Clinical trials have focused on several combination strategies between ICIs and chemotherapy, radiotherapy, or targeted molecules for this group of patients (Table 3).

#### 5.2.1. Immunotherapy with Radiotherapy

Although the MSI-H/dMMR tumors in the ICIs have achieved a significant and durable response, the results in pMMR tumors are disappointing [192]. However, it was hypothesized that radiotherapy (RT) combined with immunotherapy might be able to overcome primary resistance to ICIs in MSS CRC [193]. There is growing evidence that by damaging DNA and inducing tumor death using RT to a single site, it would be possible to enlarge the neoantigen repertoire and to up-regulate pro-inflammatory cytokines, thereby enhancing the immunotherapy effect. This phenomenon is described by the abscopal effect [194]. To date, the combination of immunotherapy and RT is being investigated in two clinical fields. The first one is the oligometastatic setting when locoregional RT is administered with curative intent, and immunotherapy can prevent distant and local relapse and enhance the response within the irradiation field. The second field involves the metastatic setting when RT to a metastatic site is expected to synergize with immunotherapy [195]. Duffy et al. investigated the combination of an anti-PD-1 agent (AMP224) with stereotactic body radiation directed against liver metastasis in mCRC patients. The treatment was feasible and safe; however, the preliminary results showed no objective response [196]. In a phase II study including 40 refractory pMMR mCRC patients, the administration of nivolumab + ipilimumab in combination with RT (to a single metastatic site) showed promising results in terms of efficiency and feasibility [24,197]. A different study conducted by Zhou et al. followed the response to ICI in combination with chemoradiotherapy (CRIT) of five advanced and metastatic CRC patients harboring MSI-H/dMMR. The ORR was 100%, with three patients achieving CR and two patients with PR, with acceptable toxicity. This retrospective study hints that CRIT could enhance the efficacy of anti-PD-1 immunotherapy and overcome potential resistance mechanisms [198].

#### 5.2.2. Immunotherapy with Chemotherapy

Similarly, the addition of chemotherapy to pMMR tumors could modify the immune contexture by generating immunogenic cell death, releasing neoantigens, and therefore activating an immune response against cancer cells [199]. Starting from this biological rationale, some preclinical studies have shown the role of chemotherapy in sensitizing malignant cells to ICIs in lung cancer models, supporting such studies in other malignancies [200]. A phase II study evaluated the efficacy of the FOLFOX regimen in combination with pembrolizumab in untreated mCRC, including 22 pMMR cases, 3 dMMR, and 5 patients with no available data. The results showed an ORR of 53% and a DCR of 100% at eight weeks [201].

Moreover, temozolomide (TMZ) is an oral alkylating agent that can generate a high number of somatic mutations in cancer cells, and therefore induce an MSI-phenotype in pMMR mCRC. TMZ methylates DNA strands, inhibits replication, and induces apoptosis [202]. The efficacy of TMZ is reduced by the O6-methylguanine methyltransferase enzyme (MGMT), which is coded by the MGMT gene. Therefore, silencing the MGMT gene could enhance the sensitivity of tumor cells to TMZ [203]. The MAYA trial (NCT03832621) evaluates the efficacy of nivolumab, ipilimumab, and TMZ in pMMR and MGMT-silenced mCRC patients, who did not progress following two cycles of TMZ [204]. The ARETHUSA trial (NCT003519412) is a phase II non-randomized study in which dMMR mCRC patients are treated with pembrolizumab until disease progression; moreover, the mCRC patients with dMMR, RAS-mutated, and MGMT IHC-negative/promoter hypermethylation positive are treated with TMZ until disease progression. By the time of progression, a tumor biopsy is performed to determine TMB. If it is >20 mutations/Mb, the patients receive pembrolizumab. The study’s primary endpoint is ORR in pMMR mCRC patients who received pembrolizumab [205].

#### 5.2.3. Immunotherapy with Chemotherapy and Targeted Agents

There are accumulating pieces of evidence that anti-vascular endothelial growth factor (VEGF) monoclonal antibody and bevacizumab could have immunomodulatory properties. It is known that VEGF can trigger T regulatory cell proliferation, increase MDSCs infiltration in the TME, and it can also favor CTLs exhaustion by the upregulation of the suppressive immune checkpoint molecule [206]. Therefore, it is a compelling rationale for the association of anti-VEGF with ICIs [207,208]. From this perspective, a clinical study evaluated the activity of the combination between bevacizumab and anti-PD-L1 atezolizumab (cohort A), or the same combination associated with modified FOLFOX6 chemotherapy (cohort B) in mCRC patients. In cohort A, one patient achieved partial response (ORR 1/14), and nine patients had stable disease. In cohort B, the ORR was 52% (12/23), and a median PFS was 14.1 months. The phase Ib REGONUVO trial assessed the efficacy and safety of regorafenib (80–160 mg/day) and nivolumab (3 mg/kg) in metastatic gastric cancer and mCRC. The study included 25 mCRC, from whom 24 (96%) cases had MSS/pMMR phenotype. The results showed an ORR of 36% in the mCRC cohort. The median PFS was 7.9 months, and the median OS was not reached [209]. These preliminary results suggesting a potential synergistic activity must be confirmed in more extensive randomized trials [210].

Moreover, early phase trials are ongoing using ICIs and antiangiogenic molecules in mCRC: NCT03396926 (pembrolizumab + capecitabine + bevacizumab), NCT03081494 (anti-PD-1 inhibitor (PDR001) + regorafenib), and NCT02848443 (nivolumab + TAS-2+oxaliplatin + bevacizumab).

#### 5.2.4. Immunotherapy with MEK Inhibition

MEK is an essential signaling molecule in the MARK pathway. Preclinical and clinical trials suggested that the inhibition of MEK in association with ICIs could up-regulate MHC class I and increase CD8+ infiltration in the tumor microenvironment, thereby generating a more effective antitumor activity [211,212]. Promising results came from an early phase I clinical trial in chemo-refractory mCRC patients evaluating the combination of the MEK inhibitor Cobimetinib with Atezolizumab, with an ORR of 17% (4/21). However, a confirmatory phase III clinical study investigating the use of atezolizumab with or without cobimetinib failed to replicate the clinical benefit over regorafenib (a multi-tyrosine kinase inhibitor) in patients with chemo-refractory MSI-L/pMMR mCRC (Table 3). The CheckMate 9N9 phase 1/2 trial is currently evaluating the efficacy and safety of nivolumab ± ipilimumab in combination with tremelimumab (MEK inhibitor) in recurrent mCRC patients (NCT03377361).

#### 5.2.5. Immunotherapy with Colony-Stimulating Factor 1 Receptor

MDSCs represent a heterogeneous population of relatively immature myeloid cells demonstrated to display a powerful immunosuppressive activity in numerous solid tumors, including CRC [213,214,215]. Colony-stimulating Factor 1 Receptor (CSF1R) is present on the monocyte surface, and its activation by the colony-stimulating factor (CSF) could promote the differentiation in MDSCc. It was hypothesized that inhibiting CSF1R will lead to suppression of MDSCc, and therefore delayed tumor growth [213]. A phase I clinical study including patients with pancreatic and CRC found the association between durvalumab (anti-PD-L1) and pexidartinib (CSF1 R inhibitor) to have an acceptable toxicity profile with no unexpected events. Further clinical data are expected in this regard [216].

#### 5.2.6. Immunotherapy with Carcinoembryonic Antigen T-Cell Bispecific

Carcinoembryonic antigen (CEA), part of the immunoglobulin supergene family, is overexpressed in most mCRC. CEA CD3 T-cell bispecific (TCB) represents a TCB antibody that can bind simultaneously at CD3 of T-cells and CEA on tumor cells. Tabernero et al. evaluated the efficacy of CEA TCB alone or combined with atezolizumab in chemo-refractory mCRC. In the monotherapy cohort (31 patients), two (6%) patients had a partial response, and the disease control rate was 45%. In the second cohort (11 patients), seven (64%) patients had stable disease, and two (18%) cases had a partial response. However, CEA TCB treatment had a complex safety profile which might be an issue for its future development [217].

## 6. Conclusions

Despite notable improvements in the diagnosis and treatment of CRC patients, the metastatic disease still has a poor prognosis with a median OS of 30 months. In recent years, we have witnessed the remarkable impact generated by immunotherapy in selected tumors. To date, most digestive tumors benefit very little from these therapeutic strategies. MSI-H/dMMR mCRC patients demonstrated an objective and sustained response to immunotherapy. However, given the heterogeneity of tumors and environmental conditions, even in this selected subset of patients, intrinsic and acquired resistance was described. Furthermore, predictive, and prognostic biomarkers and genetic alterations that could impair the efficacy of ICIs are still suboptimal. The current evidence regarding the response to ICIs suggests that predictive models would be more helpful than single biomarkers. Moreover, to enhance the efficacy of immunotherapy, we need an improved phenotypic description of the immune cells and a comprehensive understanding of the TME.

Future technological progress is expected to deepen our knowledge of the immune system by focusing on the entire genome, detecting new immune cells with clinical relevance, and developing new approaches to target cancer cells precisely. Further insight into innate and acquired resistance will lead to optimal combinatorial strategies to counteract immune escape.

## Figures and Tables

**Figure 1 life-12-00229-f001:**
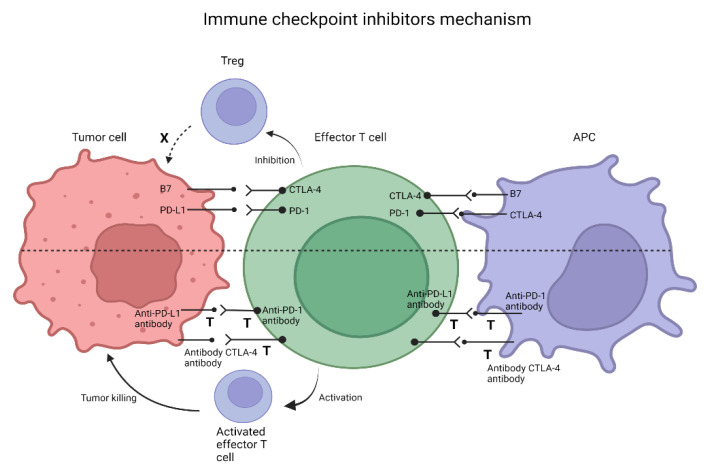
Mechanism of immunotherapy checkpoint inhibitors. When PD-1 located on the surface of effectors T cells interacts with PD-L1 on the surface of tumor cells, downstream signaling pathways are activated, inhibiting apoptosis, and promoting the conversion of effector T-cells to Tregs. CTLA-4 on the surface of T-cells can bind preferentially to the receptors (B7-1; B7-2) on the surface of APC to limit T-cell activity and proliferation in a similar way.

**Figure 2 life-12-00229-f002:**
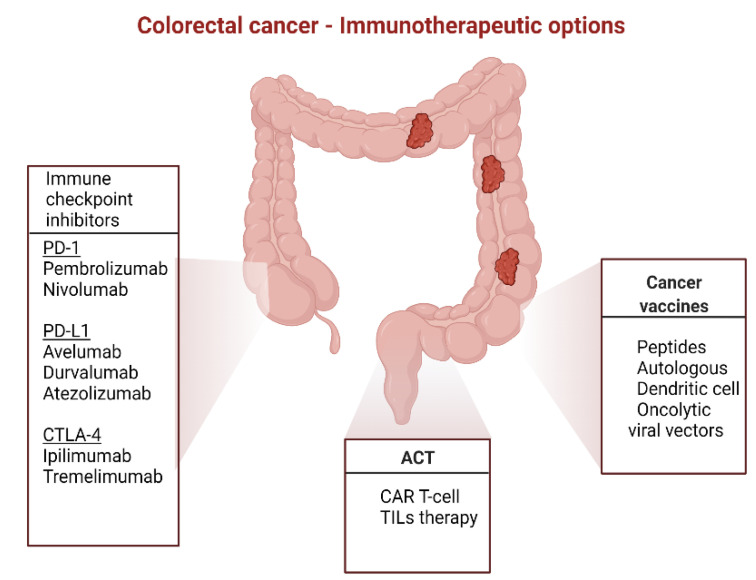
Current immunotherapeutic options in CRC.

**Table 1 life-12-00229-t001:** Ongoing clinical trials for MSI-H/dMMR CRC.

Study Name	Status	Phase	Study Population	Treatment	Endpoint	Purpose
NCT02982694	Recruiting	II	Advanced chemotherapy resistant MSI-like CRC	Atezolizumab + bevacizumab	ORR	To determine the anti-tumor effect of atezolizumab in combination with bevacizumab in chemotherapy-resistant MSI-H/dMMR CRC
NCT02997228	Recruiting	III	MSI-H/dMMR mCRC	Atezolizumab vs. atezolizumab + bevacizumab + FOLFOX	PFS	To compare mFOLFOX6/bevacizumab/atezolizumab with atezolizumab alone as the first-line treatment in MSI-H/dMMR mCRC
NCT04014530	Recruiting	I-II	dMMR and pMMR mCRC and dMMR endometrial carcinoma	Pembrolizumab + Ataluren	AE and maximum tolerable dose of Ataluren AE of the combination ORR	Efficacy of pembrolizumab in combination with Alaturen in pMMR/dMMR mCRC and dMMR metastatic endometrial carcinoma
NCT03638297	Recruiting	II	MSI-H/dMMR CRC	Pembrolizumab + COX inhibitor (aspirin)	RR	Safety and efficacy of pembrolizumab in combination with COX inhibitor in MSI-H/dMMR or high TMB CRC
NCT04001101	Recruiting	II	MSI-H/dMMRmetastatic solid tumors	Pembrolizumab + RT (metastatic site) vs. pembrolizumab	ORR	To determine if the ORR is improved by the addition of radiotherapy to pembrolizumab in MSI-H/dMMR metastatic solid tumors, compared to pembrolizumab alone
NCT04730544	Recruiting	II	MSI-H/dMMR mCRC	Nivolumab + ipilimumab	AE PFS	To determine the safety and efficacy of two combination regiments of nivolumab + opilimumab in MSI-H/dMMR mCRC
NCT04008030	Recruiting	III	MSI-H/dMMR mCRC	Nivolumab vs. nivolumab + ipilimumab Nivolumab + ipilimumab vs. chemotherapy	PFS	To compare the clinical benefit of nivolumab alone, nivolumab + ipilimumab or investigator’s choice chemotherapy in MSI-H/dMMR mCRC
NCT03104439	Recruiting	II	MSI-H/dMMR CRC, MMS CRC, pancreatic cancer	Nivolumab + ipilimumab + RT	DCR	To evaluate the combination of nivolumab, ipilimumab, and radiation therapy in MSS/MSI-H/dMMR CRC and pancreatic cancer
NCT02060188	Active, not recruiting	II	Recurrent or metastatic MSI-H and non-MSI-H CRC	Nivolumab Nivolumab + ipilimumab Nivolumab + ipiliumab + cobimetinib Nivolumab + BMS-986016 Nivolumab + daratumumab	ORR	To evaluate nivolumab alone or in combination with other anti-cancer molecules in recurrent or metastatic MSI-H or non-MSI-H CRC
NCT03186326	Recruiting	II	MSI-H/dMMR mCRC	Avelumab	PFS	Tolerance and effectiveness of Avelumab, compared to the second line standard chemotherapy for MSI-H/dMMR mCRC
NCT03475953	Recruiting	I-II	Advanced or metastatic solid tumors, including MSI-H/dMMR CRC	Avelumab + regorafenib	RP2D for regorafenib ORR PFS	To evaluate efficacy and safety of regorafenib in combination with avelumab in advanced/metastatic solid tumors
NCT03435107	Active, not recruiting	II	MSI-H/dMMR or POLE mutated mCRC	Durvalumab	ORR	To investigate durvalumab in previously treated MSI-H/dMMR or POLE mutated mCRC
NCT02983578	Active, not recruiting	II	Advanced pancreatic cancer NSCLC dMMR CRC	Danvatirsen+durvalumab	AEs, SAEs	To evaluate danvatirsen and durvalumab in patients with advanced pancreatic cancer, NSCLC, and dMMR CRC refractory to standard therapy

**Table 2 life-12-00229-t002:** Completed clinical trials for MSI-H/dMMR CRC.

Study Name	Phase	Study Population	Treatment	Primary Endpoint	Results	Purpose
NCT02460198	II	Previously treated LA unresectable or mCRC MSI-H/dMMR	Cohort A: pembrolizumab after ≥2 prior lines of therapy Cohort B: pembrolizumab after ≥1 prior line of therapy	ORR	OR = 33%/33%	To determine the efficacy of pembrolizumab monotherapy in previously treated LA unresectable or mCRC MSI-H/dMMR patients
NCT01876511	II	MSI tumors (Cohort A: MSI + CRC; Cohort B: MSI − CRC; Cohort C: MSI + non-CRC)	Pembrolizumab	irPFS (A,B), irORR (A,B), irPFS (C), ORR (A,C), PFS (A,C)	IrORR A = 40%, irPFS A = 78%; irORR B = 0%, irPFS B = 11%, Median PFS A = not reached; Median OS A = not reached; Median PFS B = 2.2 months; Median OS B = 5 months; irORR C = 71%, irPFS = 67%	To determine the anti-tumoral activity of pembrolizumab in MSI/MSS cohorts
NCT02178722	I/II	Selected cancers (including MSI-H CRC)	Pembrolizumab + epacadosat	I: TEAE; II: ORR	Acceptable safety profile ORR CRC = N/A	To assess the safety, tolerability, and efficacy of combination therapy pembrolizumab + epacadosat in patients with certain cancers.
NCT02335918	I II	Advanced refractory solid tumors (including CRC)	Varlilumab + nivolumab	I: TEAE II: ORR	Acceptable safety profile PR = 5% CRC SD = 17% CRC	To determine the clinical benefit, safety, and tolerability of combination therapy between varlilumumab + nivolumab in certain advanced refractory solid tumors.
NCT02227667	II	Advanced MSI-H CRC	Durvalumab	ORR	22%	To determine the effects of durvalumab therapy in advanced MSI-H CRC patients.
NCT02777710	I	Metastatic/ advanced CRC and PaC	Durvalumab + pexidartinib	1.DLT 2.ORR	Acceptable safety profile ORR (2 m) = 21%	To evaluate the safety and activity of durvalumab combined with pexidartinib in patients with metastatic/advanced pancreatic or CRC

**Table 3 life-12-00229-t003:** Completed clinical trials investigating immunotherapy in MMS/pMMR CRC.

Study Name	Phase	Study Population	Treatment	Primary Endpoint	Results	Purpose
NCT02981524	II	Advanced pMMR CRC	Pembrolizumab+ cyclophosphamide+ Colon cancer vaccine	ORR	No OR with DCR = 18%	To assess the efficacy (as measured by RECIST criteria) of therapy with CY/GVAX in combination with pembrolizumab in patients with advanced pMMR CRC
NCT03274804	I	Refractory MSS/ pMMR mCRC	Pembrolizumab + Maraviroc	Feasibility rate of the combined therapy	FR = 94.7%	To determine the feasibility rate of combination therapy between pembrolizumab and maraviroc in previously treated subjects who have refractory MSS/pMMR mCRC
NCT02860546	II	MSS CRC	Nivolumab + tipiracil hydrochloride	irORR	No tumor response	To evaluate the efficacy of nivolumab + tipiracil hydrochloride in patients with MSS refractory mCRC
NCT03258398	II	MSS CRC	Avelumab + tomivosertib vs. tomivosertib	Part 1: DLT during the first treatment cycle Part 2: ORR	Part 1: Acceptable safety profile for combination therapy Part 2: N/A	To evaluate the safety, tolerability, and anti-tumor activity of tomivosertib with or without avelumab in MSS CRC patients
NCT02811497	II	Advanced solid tumors (including MSS CRC)	Azacitidine + durvalumab	ORR	No OR with DCR = 7.1 and median PFS = 1.9 m and OS = 5 m	To assess the antitumor activity of azacitidine in combination with druvalumab in advanced solid tumors
NCT03005002	I	MSS mCRC (Liver)	Durvalumab + tremelimumab following radioembolization (RE) with SIR-spheres	Safety and hepatic response rate	Safety of RE followed by D + T Lack of clinical response	To determine the safety and the hepatic response rate of durvalumab+tremelimuma following RE in MSS CRC that has spread to the liver
NCT02876224	Ib	Non MSI-H mCRC	Cobimetinib + Bevacizumab + atezolizumab	TEAE	Acceptable safety profile and manageable AEs	To assess the safety, tolerability, and pharmacokinetics of oral cobimetinib with IV atezolizumab and bevacizumab in previously treated mCRC with non-MSI-H
NCT02260440	II	Chemo-refractory MSS mCRC	Pembrolizumab + azacitidine	ORR	OR = 3%	To evaluate the anti-tumor activity, safety, and tolerability of pembrolizumab in combination with azacitidine in subjects with chemo-refractory MSS mCRC
NCT03168139	I/II	MSS mCRC mPaC	Olaptesed pegol vs. olaptesed pegol + pembrolizumab	Pharmaco-dynamics Safety and tolerability	Induction of immune response and acceptable safety profile	To explore safety, tolerability, and efficacy of olaptesed monotherapy or in combination with pembrolizumab in patients with MSS mCRC and pancreatic cancer
NCT02788279	III	mCRC	Atezolizumab (A) vs. atezolizumab (A)+ cobimetinib (C) vs. regorafenib (R)	OS	OS (A) = 7.10 m OS (A + C) = 8.87 m OS (R) = 8.51 m	To compare regorafenib to cobimetinib + atezolizumab and atezolizumab monotherapy in the setting of mCRC

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
