# Peer review of "Landscape of Immunotherapy Options for Colorectal Cancer: Current Knowledge and Future Perspectives beyond Immune Checkpoint Blockade"

_life, 2022, doi:10.3390/life12020229_

Round 1

Reviewer 1 Report

The manuscript is descriptive and informative. I might suggest Section 2.6 is not necessary included, since it is not related to the main theme and no further discussion on this part in the following paragraphs.  Instead, the mechanism and future strategies for immunotherapy resistance should be more addressed.

Author Response

Manuscript ID: life-1536041

Title: Landscape of Immunotherapy Options for Colorectal Cancer: Current 
Knowledge and Future Perspectives beyond Immune Checkpoint Blockade

Authors: Alecsandra Gorzo, Diana Galos, Simona Ruxandra Volovat, Cristian 
Virgil Lungulescu *, Claudia Burz, Daniel Sur *

Reviewer 1 comments: 
The manuscript is descriptive and informative. I might suggest Section 2.6 is not necessary included, since it is not related to the main theme and no further discussion on this part in the following paragraphs.  Instead, the mechanism and future strategies for immunotherapy resistance should be more addressed.

We want to thank the reviewer for the time assigned to analyze our manuscript. We are confident that we will improve the current paper by answering the reviewer’s requests. Our primary purpose is to present concise data regarding the landscape of immunotherapy and updated therapeutic options in colorectal cancer to help clinicians and researchers in the field.
We have amended our manuscript according to the reviewer’s suggestions. The changes were done using track changes. Therefore, we deleted section 2.6 regarding Microbiota and discussed more extensively the mechanism and future strategies for immunotherapy resistance. Now the text can be read like:
“In patients who developed resistance to PD-1 blockade, the whole-exome sequencing of the tumors showed mutations in Janus kinases 1 and 2 (JAK1 and JAK2)(161). Truncating mutations in JAK 1/2 were correlated to a lack of IFN-γ responsiveness in cancer cells and, consequently, with secondary resistance to ICIs (162,163). Additionally, in JAK mutated MSI CRCs, melanoma, endometrial cancer, the expression of the PD-L1 gene was significantly down-regulated(164,165). In melanoma cells, it was shown that the IFN-y signaling pathway regulates the expression of PD-L1 through JAK1/2. Therefore, cancer cells might evade IFN-y-immune response throughout JAK1/2 mutations, leading to impaired IFN-y signaling and preventing PD-L1 expression(166). Signal transducer and activator of transcription proteins (STATs 1/2), members of this pathway, which function downstream of JAK signaling, are potent mediators of IFN-γ. Mutations in STAT proteins resulting in loss of function could generate impaired IFN-γ signaling and, therefore, immune escape (167). 
It was recently discovered that the Wnt/β-catenin pathway coordinates the tumor microenvironment and immune cell infiltration(168). A preclinical study using murine melanoma models demonstrated that increased expression of the Wnt/β-catenin pathway could decrease IFN-γ levels and T-cell function as a consequence(169). Another study on melanoma cells also showed that hyperactivation of the Wnt/β-catenin pathway reduces T-cell infiltration in the TME, leading to reduced efficacy of ICIs(170). Regarding MSI-H/dMMR CRC patients, the WNT signaling pathway should be further analyzed in the context of ICIs responsiveness. “

“In order to extend the curative potential of cancer immunotherapies, novel delivery systems are needed. Ongoing research investigates various delivery platforms like implants, nanoparticles, biomaterials, and scaffolds (186). Among their many benefits, we can mention the following: protecting and keeping the cargo inactive until it accumulates in the targeted cells, allowing localized and controlled drug delivery to minimize toxicities (187). For example, to reduce the side effects following systemic administration, ICIs were linked to a peptide derived from PLGF2 (placental growth factor 2) with a good affinity for numerous matrix proteins. In melanoma and breast cancer models, these conjugates remained more localized near the tumor site after peritumoral administration (188). Moreover, new delivery platforms, like nanoparticles, could attenuate drug exposure of particular tissues caused by therapeutic combinations (chemotherapy and immunotherapy) that would otherwise be toxic for the patient (187,189). Besides the many benefits already mentioned, new delivery technologies could address the limitations set by resistance mechanisms. For instance, delivery systems could be expanded to modulate immunogenicity in tumors with cold microenvironments and enhance the response to ICIs (190). As immunotherapy is evolving very fast, all the advances made in drug delivery will significantly contribute to personalized medicine.”

To conclude, we want to address our gratitude for reviewing our article.

Reviewer 2 Report

A really great review of the current options and pitfalls, clear to non oncologist also. Objectively, nothing to add to the manuscript.

Author Response

Manuscript ID: life-1536041

Title: Landscape of Immunotherapy Options for Colorectal Cancer: Current 
Knowledge and Future Perspectives beyond Immune Checkpoint Blockade

Authors: Alecsandra Gorzo, Diana Galos, Simona Ruxandra Volovat, Cristian 
Virgil Lungulescu *, Claudia Burz, Daniel Sur *

Reviewer 2 comments:

A really great review of the current options and pitfalls, clear to non oncologist also. Objectively, nothing to add to the manuscript.

We want to thank the reviewer for the time assigned to analyze our manuscript. Our primary purpose is to present concise data regarding the landscape of immunotherapy and updated therapeutic options in colorectal cancer to help clinicians and researchers. We are pleased to know that the content of our paper is received as significant addition to the scientific field.

To conclude, we want to address our gratitude for reviewing our article.

Reviewer 3 Report

  • Authors have discussed micro satellites and their instabilities, but I wonder whether authors have provided the proper insight into the available literature, since there are various studies describing various ethnic groups, type of CRCs in different populations, I think authors should also look into MSIs based on the different populations, viz, European, Asian (e.g, China and India in particular), and others.
  • Various studies describing MSI,viz; Malhotra P et al, 2014 etc, those studies should be discussed.
  • There are few more, like Chengjing Zhou et al, 2021.This should also be discussed.
  • Chemotherapy related issues should be discussed
  • Since POLE and POLD1 has very important role in familial cancer, please discuss whether this is context dependent, or it may be a general case.
  • How Immuno-score could be used as a variance for the cancer immunotherapy.
  • How prognostic ability, which is dependent on the immunoscore tends to be the classical gold-standard for TNM system in predicting DFS and OS for different stages of the disease.
  • How mono-clonal antibodies could be modified and to be delivered to be used as specific immunotherapeutic.

Author Response

Manuscript ID: life-1536041

Title: Landscape of Immunotherapy Options for Colorectal Cancer: Current 
Knowledge and Future Perspectives beyond Immune Checkpoint Blockade

Authors: Alecsandra Gorzo, Diana Galos, Simona Ruxandra Volovat, Cristian 
Virgil Lungulescu *, Claudia Burz, Daniel Sur *

Reviewer 3 comments:

ï‚·  Authors have discussed micro satellites and their instabilities, but I wonder whether authors have provided the proper insight into the available literature, since there are various studies describing various ethnic groups, type of CRCs in different populations, I think authors should also look into MSIs based on the different populations, viz, European, Asian (e.g, China and India in particular), and others.

We want to thank the reviewer for the time assigned to analyze our manuscript. We are confident that we will improve the current paper by answering the reviewer’s requests. Our primary purpose is to present concise data regarding the landscape of immunotherapy and updated therapeutic options in colorectal cancer to help clinicians and researchers in the field.  The changes were done using track changes.We have addressed the issue of CRC prevalence in different ethnic groups and MSI phenotype frequency in distinct populations by citing various papers exploring the subject. By doing so, we offer the reader an introduction to the subject and provide them with references towards extensive research on epidemiological aspects of CRC and microsatellite instability. The new phrases can be read as: “Even if high-income nations show a greater incidence in CRC, less developed countries are encountering significant increase of CRC cases (2). Extensive studies have observed differences in risk factors, incidence and cancer-related deaths between ethnic groups, with African Americans, in particular, showing a higher frequency of CRC cases as well as death rates (3-5)”

“Populations based studies have investigated the susceptibility of MSI-H/dMMR mutations in certain ethnic groups presenting with CRC. Studies show the MSI-H/dMMR phenotype has a significantly higher prevalence in the African American (AA) population (up to 45%) (38-41), while the Caucasian and Asian populations show a lower incidence of MSI-H rates, no higher than 20% (41,42).”

ï‚·  Various studies describing MSI,viz; Malhotra P et al, 2014 etc, those studies should be discussed.

We appreciate the comment of the reviewer. We have addressed the subject of MSI-H/dMMR CRC prevalence in the Indian population by addressing a significant study in the field, as previously mentioned. It can be found in the manuscript as: “ A study conducted on an Indian cohort observed similar frequencies of MSI-H/dMMR CRC in their studied population when compared to the West, despite having an inferior incidence of CRC cases (43).”

ï‚·  There are few more, like Chengjing Zhou et al, 2021.This should also be discussed.

We are thankful for the suggestion to look into the work of C Zhou et al. as we consider it valuable for the topic of our manuscript. We have addressed the study "Good Tumor Response to Chemoradioimmunotherapy in dMMR/MSI-H Advanced Colorectal Cancer: A Case Series" of C Zhou et al. in Section 5.2.1 of our paper. Now the text can be read as: “A different study conducted by Zhou et al followed the response to ICI in combination with chemoradiotherapy (CRIT) of five advanced and metastatic CRC patients harboring MSI-H/dMMR. The ORR was 100%, with three patients achieving CR and two patients having PR, with acceptable toxicity. This retrospective study hints that CRIT could enhance the efficacy of anti-PD-1 immunotherapy and overcome potential resistance mechanisms (197).”

ï‚·  Chemotherapy related issues should be discussed

We thank the reviewer for the suggestion. We want to underline that we allocated an entire section (5.2.2) for the combination between immunotherapy and chemotherapy. Further, we amended our manuscript according to your suggestion and discussed chemotherapy-related issues. The phrase can be read as: " Moreover, new delivery platforms, like nanoparticles, could attenuate drug exposure of particular tissues caused by therapeutic combinations (chemotherapy and immunotherapy) that would otherwise be toxic for the patient (187,189)”

ï‚·  Since POLE and POLD1 has very important role in familial cancer, please discuss whether this is context dependent, or it may be a general case.

We want to thank the reviewer for the suggestion of improving our manuscript. According to your input, we have amended the " Pold1/Pole " section to underline the important role of Pold1/Pole mutations in familial cancer. Now the phrase can be read like: “Germline mutations in the exonuclease domain of POLD1 and POLE affect the proofreading abilities of these polymerases, predispose to multiple colorectal carcinomas and adenomas, and generate polymerase proofreading–associated polyposis (PPAP) (70). PPAP represents 0.1-0.4% of familial cancer cases (71). Moreover, other extracolonic tumors were described, including brain, endometrial, ovarian, breast, skin tumors (72).”

ï‚·  How Immuno-score could be used as a variance for the cancer immunotherapy.

ï‚·  How prognostic ability, which is dependent on the immunoscore tends to be the classical gold-standard for TNM system in predicting DFS and OS for different stages of the disease.

We want to thank the reviewer for the recommendations. We have amended the manuscript according to your input, and we addressed more in-depth the topic regarding Immunoscore. Now the phrase can be read like this: “A high immunoscore was associated with the highest DFS and OS in stage II colon cancer. The 5-year recurrence rate was 8% in high, 14% in intermediate, and 23% in low immunoscore. A multivariable analysis had similar results (p < 0.0001 for high Immunoscore vs. low) (63). Regarding stage III colon cancer, according to the NCCTG NO157 trial, a high immunoscore was correlated with a longer 3-year DFS compared to a low immunoscore (p<0.05) (64). In the phase III IDEA trial, high and intermediate immunoscore significantly predict a DFS benefit of prolonged adjuvant chemotherapy with FOLFOX regimen in stage III colon cancer patients. (HR = 0.53; 95% CI 0.37-0.75; P = 0.0004) (65). Apart from representing a good prognostic marker, the immune contexture could also predict the response to ICIs (66). The CD8+ T cells density ware directly correlated with the clinical response to anti-PD1 agents. Moreover, CD8+ T cells were also suggested to be a good predictor of the response to anti CTLA4 molecules in melanoma patients (67).”

ï‚·  How mono-clonal antibodies could be modified and to be delivered to be used as specific immunotherapeutic.

We want to thank the reviewer for the suggestion. Since immunotherapy shifted the paradigm of cancer treatments, manufacturing and delivery technologies are a true challenge. We have included in our manuscript valuable information regarding this topic. Now the phrase can be read like this:

“In order to extend the curative potential of cancer immunotherapies, novel delivery systems are needed. Ongoing research investigates various delivery platforms like implants, nanoparticles, biomaterials, and scaffolds (186). Among their many benefits, we can mention the following: protecting and keeping the cargo inactive until it accumulates in the targeted cells, allowing localized and controlled drug delivery to minimize toxicities (187). For example, to reduce the side effects following systemic administration, ICIs were linked to a peptide derived from PLGF2 (placental growth factor 2) with a good affinity for numerous matrix proteins. In melanoma and breast cancer models, these conjugates remained more localized near the tumor site after peritumoral administration (188). Moreover, new delivery platforms, like nanoparticles, could attenuate drug exposure of particular tissues caused by therapeutic combinations (chemotherapy and immunotherapy) that would otherwise be toxic for the patient (187,189). Besides the many benefits already mentioned, new delivery technologies could address the limitations set by resistance mechanisms. For instance, delivery systems could be expanded to modulate immunogenicity in tumors with cold microenvironments and enhance the response to ICIs (190). As immunotherapy is evolving very fast, all the advances made in drug delivery will significantly contribute to personalized medicine.”

.”

To conclude, we want to address our gratitude for reviewing our article.

Round 2

Reviewer 3 Report

The revised manuscript entitled “Landscape of Immunotherapy Options for Colorectal Cancer: Current Knowledge and Future Perspectives beyond Immune Checkpoint Blockade” by Alecsandra Gorzo et al, describes and reviewed the existing data in context to immunotherapy in CRC, and more accurately in a subset of patients with MSI-H tumors. The authors addressed the importance of biomarkers in selecting CRC patients for immunotherapy and also presented the challenges which come into play due to the resistance mechanisms and some potential future strategies which can be utilized for immunotherapy uses.

The authors have included some of the new studies in the manuscript based on the available literature, but there remain certain sections for example, prognostic Biomarkers for Selecting CRC that needs the addition of literature and the inclusion of the studies which were mentioned by the reviewers.